# TOWARDS REALISTIC HYPERPARAMETER OPTIMIZATION IN CONTINUAL LEARNING

## ABSTRACT

In continual learning (CL)—where a learner trains on a stream of data—standard hyperparameter optimisation (HPO) cannot be applied, as a learner does not have access to all of the data at the same time. This has prompted the development of CL-specific HPO frameworks. The most popular way to tune hyperparameters in CL is to repeatedly train over the whole data stream with different hyperparameter settings. However, this *end-of-training* HPO is unrealistic as in practice a learner can only see the stream once. Hence, there is an open question: *what HPO framework should a practitioner use for a CL problem in reality?* This paper answers this question by comparing several realistic HPO frameworks. We find that none of the HPO frameworks considered, including end-of-training HPO, perform consistently better than the rest on popular CL benchmarks. We therefore arrive at a twofold conclusion: a) on the popular CL benchmarks examined, a CL practitioner should select the HPO framework based on other factors, for example compute efficiency and b) to be able to discriminate between HPO frameworks there is a need to move beyond the current most commonly used CL benchmarks.

## 1 INTRODUCTION

Sequentially updating deep learning systems on a non-stationary data stream is a challenging problem which *continual learning* (CL) methods aim to address. The standard CL setup is when a learner sees a sequence of tasks one-by-one and at the end of learning is evaluated on how well it performs across all tasks. There have been many methods (Delange et al., 2021; Parisi et al., 2019; Wang et al., 2023) designed for this problem and CL scenarios proposed (Hsu et al., 2018; Antoniou et al., 2020; van de Ven & Tolias, 2019). A key decision when using a CL method is selecting hyperparameter settings—learning rates, regularisation coefficients, etc. (Feurer & Hutter, 2019; Delange et al., 2021; Wistuba et al., 2023). The most common way to fit hyperparameters for CL is *end-of-training* hyperparameter optimisation (HPO) (Delange et al., 2021; Buzzega et al., 2020)—shown in Figure 1. This is when the hyperparameters are fit by training over the whole data stream with each hyperparameter configuration and then selecting the configuration that has the best end-of-training performance on a held-out validation set. However, end-of-training HPO is unrealistic as in the real world a learner can only train over the data stream once and must select hyperparameters only using the data it can currently access. Therefore, determining the best *realistic* way to perform HPO for CL is currently an open problem.

In this work, we address the problem of deciding what realistic HPO framework to use in CL. To do this, we benchmark a variety of approaches for performing HPO across different CL methodologies (ER (Chaudhry et al., 2020), ER-ACE (Caccia et al., 2021), iCaRL (Rebuffi et al., 2017), ESMER (Sarfraz et al., 2023) and DER++ (Buzzega et al., 2020)). We investigate both fixed HPO frameworks where the hyperparameters are kept constant throughout training and dynamic HPO frameworks where hyperparameters are adapted throughout learning. For fixed HPO we examine (i) *end-of-training* HPO as well as (ii) a *first-task* HPO framework where we fit the hyperparameters only using data from the first task (see Figure 1), a realistic and computationally efficient method. For dynamic HPO, we consider (i) using data from the current task, (ii) using data stored in memory, and (iii) using validation sets from previous tasks to perform HPO for each new task. By comparing these different HPO frameworks we shed light on what validation signal is sufficient to fit hyperparameters in CL and whether hyperparameters need to be adapted during training.

Figure 1: Depiction of end-of-training and first-task HPO frameworks, which fix the hyperparameters (HPs) throughout training. *End-of-training* HPO is the most common HPO framework for CL and works by training over the whole data stream for each HP configuration and then uses a validation set consisting of data from each task to select the best HPs. End-of-training HPO is unrealistic as it assumes you have access to all of the data stream from the start of training. On the other hand, *first-task* HPO selects HPs by repeatedly training and validating performance on the first task, which can be used in the real world and is more efficient.

Our experiments show that all the HPO frameworks tested perform similarly in terms of predictive performance; no one method is consistently better than the others. This suggests that on the popular CL benchmarks used in our experiments other factors should be used to select the HPO framework. For example, if a CL researcher using these benchmarks wanted to maintain performance while reducing compute cost of HPO, they could use the realistic and most computationally efficient method, *first-task* HPO. Additionally, it suggests that future work on HPO in CL should move beyond the use of these standard benchmarks to ones where there is likely a performance difference between HPO frameworks.

The main contributions of this work are:

- We benchmark a suite of realistic CL HPO frameworks against the commonly used but unrealistic end-of-training HPO. This is, to the best of our knowledge, the first comprehensive comparison across several HPO frameworks for CL.

- We show that all HPO frameworks we compare perform similarly in our experiments. This suggests that, on the benchmarks looked at, there are several realistic HPO frameworks which can be used instead of the commonly used but unrealistic end-of-training HPO framework.

- We provide evidence for common CL benchmarks that—as the predictive performance of HPO frameworks are similar—other factors should be used to select a realistic HPO framework. For example, to minimise compute cost first-task HPO is a good method.

- Our experiments highlight that to be able to better compare and develop CL HPO frameworks there is a need to move beyond the current most popular CL benchmarks.

## 2 PRELIMINARIES AND RELATED WORK

CL is a large research area where many different settings have been looked at. In this work we look at the most common CL setting which is known as standard CL, or sometimes offline CL (Prabhu et al., 2020). In *Standard CL*, the learner sees a non-stationary sequence of data chunks called *tasks* one-by-one, such that it only has access to one chunk at a time and cannot access previously seen or future chunks. Each task consists of examples which are data instance and label pairs (e.g. pairs of images and their class) sampled from a subset of the classes. For example, the first task might be examples of cows and sheep and the second task could be formed of examples of dogs and cats. The goal of the learner is to classify new examples accurately after training on the whole data stream. There are two common ways to evaluate a CL learner, task and class incremental learning. *Task-incremental* learning is when, at test-time, the learner knows which task a data instance comes from and so only needs to distinguish between classes within that task. While, *class-incremental* learning is when the learner is not given what task a data instance belongs to at test time and must distinguish between all classes from all the tasks. An important part of the standard CL setting is the assumption of memory constraints, which is why a learner cannot solve CL by storing previous data chunks in memory. The memory constraints take the form of only allowing a learner to store a small amount of previous data in memory and in constraining its use of memory for storing additional networks or parts of networks (Delange et al., 2021; Wang et al., 2023).

There have been many methods proposed for CL (Delange et al., 2021; Parisi et al., 2019; Wang et al., 2023). One of the most popular and performant approaches to standard CL are replay methods (Wang et al., 2023). This is especially true for class-incremental learning, where they are commonly the best performing methods (van de Ven & Tolias, 2019; Wu et al., 2022; Mirzadeh et al., 2020; Lee & Storkey, 2024). *Replay* methods use a memory buffer to store a set of examples from previous tasks to regularise the updates on new tasks such that the learner does not forget previous task knowledge. For example, the stereotypical replay method is *experience replay* (ER) (Chaudhry et al., 2020; 2019b; Aljundi et al., 2019a) which for each learning step appends a sample of data from the replay buffer to the batch of current task data to be trained on. More complex replay methods often use a form of knowledge distillation on a sample of data from the replay buffer. For example, DER++ (Buzzega et al., 2020), ESMER (Sarfraz et al., 2023) and iCaRL (Rebuffi et al., 2017) are replay methods which use a method-specific knowledge distillation term. For each of these methods the most common hyperparameters that are tuned are the learning rate and regularisation coefficients, which need to be tuned to get good performance (see Appendix B). While other potential hyperparameters are often not tuned in CL, e.g. momentum (Buzzega et al., 2020).

While the most common HPO framework used in standard CL is end-of-training HPO, there have been several other HPO frameworks suggested (Kilickaya & Vanschoren, 2023; Parisi et al., 2019; Cai et al., 2021). For example, Delange et al. (2021) propose a dynamic HPO framework. The method adapts the hyperparameters for each task by first training with the hyperparameter configuration which is assumed to have the least impact on previous task performance. Then the method incrementally changes hyperparameter values to improve performance on the current task to a pre-specified value, while decreasing performance on previous tasks. However, this method assumes that the direction to change hyperparameters to increase performance on the current task is known and that the interaction between different hyperparameters is understood. In this work we look at a similar HPO framework, current-task HPO, which does not need the above assumptions. Also, for the online CL scenario—which is different to standard CL—another HPO framework has been proposed whereby end-of-training HPO is used on the first (or first $k$) tasks and then the hyperparameters are fixed after that (Chaudhry et al., 2019a). To the best of our knowledge, this HPO framework has been rarely used in standard CL up to this point. Here, we look at it in the form of the first-task HPO framework and examine how it performs in the commonly used standard CL setting. There has also been work on making dynamic HPO frameworks more efficient by sampling fewer HPO configurations, for example using bandit methods (Liu et al., 2023) and analysis of variance techniques (Semola et al., 2024). However, for simplicity, we only look at the more expensive dynamic HPO frameworks which are an upper bound to the performance of these more efficient methods. While as shown above there has been work on HPO for CL, to the best of our knowledge there has not been a comprehensive comparison between the main HPO frameworks proposed. This is one of the key contributions of this work, shedding light on the relative performances of HPO frameworks for CL.

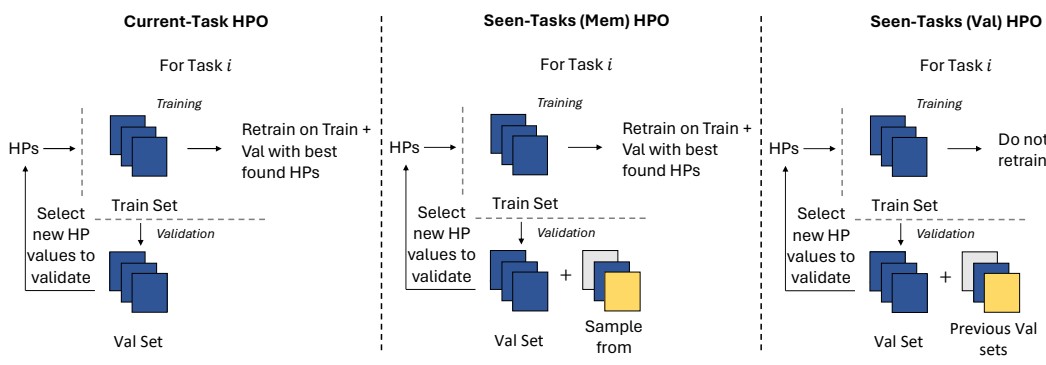

Figure 2: Depiction of current-task, seen-tasks (Mem) and seen-tasks (Val) HPO frameworks, which dynamically adapt hyperparameters (HPs) for each task. Each methods splits the data of the current task into train and validation sets. Then, current-task HPO uses this validation set to fit the HPs for the current task. While, seen-tasks (Mem) and seen-tasks (Val) use a combination of this validation set and either a sample of data from previous tasks stored in memory or validation sets of previous tasks, respectively. Then current-task and seen-tasks (Mem) HPO retrain on the combined validation and train sets to complete the learning process on that task. Seen-tasks (Val) does not retrain, instead it takes the model fitted using the best found hyperparameters as the final model for the current task. This is to ensure that the current task's validation set has not been trained on when fitting hyperparameters for future tasks.

## 3 STANDARD CL

While the setting we look at, standard CL, is mentioned above, we describe it more formally here. In *standard CL* a learner sees a sequence of tasks, $D_1, \ldots, D_T$, where each task consists of a chunk of data. The chunks of data consist of a set of examples, where an example is a pair formed of a data instance $\mathbf{x} \in X$ and label $y \in C$. Each task only contains examples from a given subset of the classes, in other words for all $(\mathbf{x}, y) \in D_i$ we have that $y \in C_i$ and $C_i \subseteq C$ is the subset of classes the examples of that task can belong to. In this work we look at the most common setting, where no two tasks have examples from the same class. This means that for any two task $i$ and $j$ we have that $C_i \cap C_j = \varnothing$. Additionally, learners can have a memory buffer of previous examples which consists at task $i$ of the set $M_i$. Training consists of the learner sequentially seeing each task in order and it cannot access the data from previous or future tasks. For each task, its data chunk is split into training and validations sets, $\text{Train}_i \subseteq D_i$ and $\text{Val}_i \subseteq D_i$, to enable the use of HPO frameworks. Then after fitting the hyperparameters the learner usually retrains on the combination of the training and validation sets, $D_i = \text{Train}_i \cup \text{Val}_i$. After training the learner is tested by evaluating its performance on a held-out set of data which consists of an equal number of examples from all the classes. We look at two evaluation scenarios, task-incremental learning and class-incremental learning. *Task-incremental* learning is where the learner receives with each test data instance the task it belongs to and therefore the subset of classes that the data instance can belong to. While for *class-incremental* learning, no indication is given of what task a test data instance belongs to.

## 4 HPO FRAMEWORKS FOR CL

In this work, we examine several HPO frameworks for CL to see which should be the preferred choice to use in CL. We look both at fixed HPO frameworks which keep the values of hyperparameters constant throughout training and dynamic HPO which adapts the hyperparameters per task. The fixed HPO frameworks we look at are end-of-training HPO and first-task HPO and the dynamic HPO frameworks we look at are current-task HPO, seen-tasks HPO (Mem) and seen-tasks HPO (Val). Each of these frameworks are described in turn below and we present an overview of their advantages and disadvantages in Table 1.

Table 1: Advantages and disadvantages of different HPO frameworks. Where, for time complexity, $K$ refers to the number of hyperparameter configurations looked at and $T$ is the number of tasks in the data stream. The asterisk (*) for seen-tasks HPO (Val) denotes that, while it does not require knowledge of future tasks like end-of-training HPO, it does require additional storage compared to other methods. The additional memory is needed to store the validation sets of previous tasks.

| HPO Framework | Realistic? | Efficient? (Time Complexity) |
|---|---|---|
| End-of-training HPO | ✗ | ✗ ($\mathcal{O}(T \times K)$) |
| First-task HPO | ✓ | ✓ ($\mathcal{O}(T + K)$) |
| Current-task HPO | ✓ | ✗ ($\mathcal{O}(T \times K)$) |
| Seen-tasks HPO (Val) | ✓* | ✗ ($\mathcal{O}(T \times K)$) |
| Seen-tasks HPO (Mem) | ✓ | ✗ ($\mathcal{O}(T \times K)$) |

**End-of-training HPO** is the most common HPO framework for CL (shown in Figure 1). It selects hyperparameters by first training each hyperparameter configuration on the *whole data stream*. Second, it evaluates the final model fitted using each hyperparameter configuration on a validation set formed of each task's held-out validation set, and selects the configuration with the highest validation performance. Last, it retrains using the selected configuration on the whole data stream where the validation data for each task is added to the training data. The model fitted at the end of this training run is the final model to be evaluated. This HPO framework is expensive as it needs to perform a training run over all the data stream for each hyperparameter configuration looked at. Additionally, it is unrealistic as it requires running through the data stream multiple times, which is not possible in many real-world settings. It might be thought that to make end-of-training more realistic the learner could store a network for each hyperparameter configuration: updating each network on every task and performing selection at the end of training. This idea would remove the requirement of running through the data stream multiple times. However, it would also require a large amount of extra memory. Additionally, the learner would have to store and not train on the validation data for each previous task. Therefore, because of underlying constraints on memory usage in standard CL, *it is not possible* to use such an idea.

**First-task HPO** is a fixed HPO framework which is illustrated in Figure 1. It selects hyperparameters by training each hyperparameter configuration on the *first task*. Next, it measures the performance of each configuration on the held-out validation set of the first task. The configuration with the highest validation accuracy is then used to retrain on the first task using both the training and validation data and thereafter for all of the future tasks. First-task HPO is computationally efficient as it trains using each hyperparameter configuration solely on the first task and then only trains using one configuration for the rest of the tasks. This is much less costly than end-of-training HPO, which for all tasks must train using each hyperparameter configuration. Additionally, first-task HPO can be used in real-world settings as it only assumes access to data available at the start of training, the first task, and not future tasks like end-of-training HPO.

**Current-task HPO** is a dynamic HPO framework which selects hyperparameters for each task using the validation set of the *current task* (shown in Figure 2). This is a greedy strategy, selecting the hyperparameters that maximise the validation performance of the current task. It is roughly as computationally expensive as end-of-training HPO, as it has to validate each hyperparameter configuration for each task. However, it is more realistic than end-of-training HPO as it only needs access to the current task's data.

**Seen-tasks HPO (Mem) and seen-tasks HPO (Val)** are dynamic HPO frameworks (shown in Figure 2). They select hyperparameters for each task by a validation set formed of current task validation data along with some historic data from the stream. We consider two ways to integrate historic task data. Seen-tasks HPO (Mem) uses a sample of data from the current memory buffer. Seen-tasks HPO (Val) uses the validation sets of previous tasks. So, unlike current-task HPO, the hyperparameters are fit using both current and previous task data. This should aid the HPO procedure in selecting hyperparameters that ensure previous tasks are not forgotten. Like current task HPO, both seen-tasks HPO (Mem) and seen-tasks HPO (Val) are as computationally expensive as end-of-training HPO. Seen-tasks HPO (Val) assumes it is possible to access the validation sets of previous tasks which makes it less realistic than current or first task HPO. This is unlike seen-tasks

HPO (Mem) which does not assume this as it uses data stored in the memory buffer to measure performance on the previous tasks. But, this comes at the cost of biasing its validation performance as the data in the memory buffer has been trained on in previous tasks.

For seen-tasks HPO (Mem), three additional details are important to mention. First, to ensure we are not training on validation data, the sample from memory used in the validation set is not trained on for the current task. Second, as the memory buffer contains different amounts of data for each task, we sample the same proportion of examples from each task to add to the validation set. Last, unlike for the other HPO frameworks, the validation set combined with the sample from memory might be class imbalanced. Therefore, unlike other methods which use validation accuracy as the performance metric, for seen-tasks HPO (Mem) we use the median of per class accuracies to reduce the impact of class imbalance.

## 5 EXPERIMENTS

**Benchmarks** In our experiments we look at two settings, the commonly used split-task setting (Buzzega et al., 2020; Delange et al., 2021) and the heterogeneous task setting. We look at these settings using the datasets CIFAR-10, CIFAR-100, CORe50 and Tiny ImageNet (Krizhevsky, 2009; Lomonaco & Maltoni, 2017; Wu et al., 2015). We chose to use these datasets and the split-task setting due to their commonplace use in the CL literature (Wang et al., 2023) and hence to maximise the insights our results can have on current practice. In the split-task setting, each task has the same number of classes associated with it and no two tasks share a class. For CIFAR-10 and CORe50, the dataset is split into five tasks, each containing the data from two or ten of the classes, respectively. For CIFAR-100 and Tiny ImageNet, the datasets are split into ten tasks, where each task contains the data of 10 or 20 classes, respectively. In the heterogeneous task setting, instead of each task having the same number of classes associated with it they have a varying amount, from two to ten, but still no two tasks share a class (see Appendix A for more details). This is to make the tasks have differing amounts of data and difficulty. We only look at CIFAR-100 and Tiny ImageNet for the heterogeneous task setting due to computational cost. Additionally, for the heterogeneous task setting we divide the datasets into twenty tasks to test how HPO frameworks perform on longer task sequences. For both settings, if required by the HPO framework, we split the data of the task into train and validations sets, where the validation set contains $10\%$ of the task's data evenly sampled from each class associated with the task.

We evaluate the methods at the end of training using a standard performance metric for CL, average accuracy (Chaudhry et al., 2019a). The average accuracy of a method is the mean accuracy over each task on a held-out test set which contains an equal amount of data from each task. For class-incremental learning, the learner must classify between all classes at test time as it is not told what task a test data instance comes from. For task-incremental learning, the learner knows what task each test data instance comes from, meaning only classes from that task will be predicted.

**CL methods** To evaluate how well each HPO framework performs we look at applying them to fit the hyperparameters of several common and well performing CL methods. More specifically, we utilise the CL methods: ER (Chaudhry et al., 2020), ER-ACE (Caccia et al., 2021), iCaRL (Rebuffi et al., 2017), ESMER (Sarfraz et al., 2023) and DER++ (Buzzega et al., 2020). For these methods we fit the learning rate and any regularisation coefficients they have using each HPO framework. While all HPO frameworks looked at can be used with any underlying sampler/selector of hyperparameter configurations, for simplicity and to be consistent with common practice in CL (Buzzega et al., 2020; Boschini et al., 2022; Sarfraz et al., 2023) we use grid search. We look at the combination of ten different learning rate values and for each regularisation coefficient three different values. This means for DER++ we search across 90 different hyperparameter configurations (learning rate and two regularisation coefficients) and for ESMER we search across 30 different configurations (learning rate and the loss margin coefficient). While, for ER, iCaRL and ER-ACE we look at 10 different configurations as they have no regularisation coefficients to fit. The hyperparameter grid used is very similar to the ones looked at in several popular works on CL (Buzzega et al., 2020; Boschini et al., 2022) and is given in full in Appendix A. Moreover, for each method we use: a ResNet18 (He et al., 2016) as the underlying backbone network; random crop and horizontal flip data augmentations when training; and a memory buffer of size 5120, in common with previous work (Buzzega et al., 2020).

Table 2: Results of using different HPO frameworks for ER, iCaRL, ER-ACE, ESMER and DER++ on the standard split-task CIFAR-10 and CIFAR-100 benchmarks. We report mean average accuracy over three runs with their standard errors and, to highlight effect size, bold results which are greater by $+0.5\%$ average accuracy than any other for that CL method. The table shows that all HPO frameworks perform similarly; none perform consistently better than the rest.

| CL Method | HPO Framework | CIFAR-10 | | CIFAR-100 | |
|---|---|---|---|---|---|
| | | Class-IL. | Task-IL. | Class-IL. | Task-IL. |
| ER | End-of-training HPO | $83.55_{\pm0.44}$ | $97.18_{\pm0.14}$ | $51.03_{\pm0.43}$ | $85.68_{\pm0.29}$ |
| | First-task HPO | $\mathbf{84.38}_{\pm0.45}$ | $96.82_{\pm0.17}$ | $49.61_{\pm0.34}$ | $84.97_{\pm0.19}$ |
| | Current-task HPO | $82.10_{\pm2.21}$ | $96.39_{\pm0.50}$ | $50.64_{\pm0.40}$ | $85.47_{\pm0.18}$ |
| | Seen-tasks HPO (Val) | $83.67_{\pm0.73}$ | $96.84_{\pm0.21}$ | $51.46_{\pm0.36}$ | $85.65_{\pm0.06}$ |
| | Seen-tasks HPO (Mem) | $79.49_{\pm0.63}$ | $95.93_{\pm0.09}$ | $47.39_{\pm0.24}$ | $84.83_{\pm0.22}$ |
| iCaRL | End-of-training HPO | $77.79_{\pm0.23}$ | $\mathbf{98.52}_{\pm0.03}$ | $54.30_{\pm0.36}$ | $85.74_{\pm0.45}$ |
| | First-task HPO | $77.83_{\pm0.22}$ | $95.31_{\pm0.12}$ | $52.56_{\pm0.10}$ | $84.60_{\pm0.09}$ |
| | Current-task HPO | $76.15_{\pm0.75}$ | $93.29_{\pm0.61}$ | $54.26_{\pm0.02}$ | $85.74_{\pm0.06}$ |
| | Seen-tasks HPO (Val) | $77.58_{\pm0.49}$ | $94.32_{\pm1.01}$ | $51.89_{\pm0.39}$ | $84.02_{\pm0.68}$ |
| | Seen-tasks HPO (Mem) | $76.67_{\pm0.44}$ | $95.41_{\pm0.28}$ | $49.16_{\pm0.23}$ | $82.43_{\pm0.23}$ |
| ER-ACE | End-of-training HPO | $82.34_{\pm0.30}$ | $96.74_{\pm0.01}$ | $55.58_{\pm0.39}$ | $85.73_{\pm0.09}$ |
| | First-task HPO | $83.20_{\pm0.79}$ | $96.67_{\pm0.18}$ | $56.36_{\pm0.29}$ | $86.11_{\pm0.154}$ |
| | Current-task HPO | $\mathbf{83.99}_{\pm0.22}$ | $96.58_{\pm0.15}$ | $56.46_{\pm0.36}$ | $86.35_{\pm0.02}$ |
| | Seen-tasks HPO (Val) | $81.94_{\pm1.55}$ | $95.90_{\pm0.51}$ | $54.37_{\pm0.25}$ | $85.02_{\pm0.14}$ |
| | Seen-tasks HPO (Mem) | $81.61_{\pm0.15}$ | $96.40_{\pm0.13}$ | $53.76_{\pm0.21}$ | $84.56_{\pm0.31}$ |
| ESMER | End-of-training HPO | $80.73_{\pm0.15}$ | $96.50_{\pm0.01}$ | $56.16_{\pm0.54}$ | $88.69_{\pm0.35}$ |
| | First-task HPO | $77.89_{\pm0.46}$ | $96.15_{\pm0.12}$ | $56.61_{\pm0.20}$ | $89.05_{\pm0.10}$ |
| | Current-task HPO | $81.69_{\pm0.25}$ | $96.03_{\pm0.05}$ | $55.11_{\pm0.13}$ | $88.96_{\pm0.08}$ |
| | Seen-tasks HPO (Val) | $81.29_{\pm0.03}$ | $96.46_{\pm0.06}$ | $53.81_{\pm0.44}$ | $87.26_{\pm0.13}$ |
| | Seen-tasks HPO (Mem) | $70.95_{\pm0.94}$ | $95.79_{\pm0.14}$ | $\mathbf{57.50}_{\pm0.14}$ | $89.27_{\pm0.16}$ |
| DER++ | End-of-training HPO | $84.40_{\pm0.94}$ | $95.75_{\pm0.33}$ | $56.04_{\pm3.67}$ | $83.13_{\pm2.69}$ |
| | First-task HPO | $85.22_{\pm0.08}$ | $96.14_{\pm0.10}$ | $55.20_{\pm0.78}$ | $81.68_{\pm0.66}$ |
| | Current-task HPO | $84.90_{\pm0.11}$ | $95.92_{\pm0.11}$ | $55.00_{\pm1.21}$ | $83.14_{\pm0.76}$ |
| | Seen-tasks HPO (Val) | $85.44_{\pm0.38}$ | $96.22_{\pm0.15}$ | $56.59_{\pm0.64}$ | $83.61_{\pm0.42}$ |
| | Seen-tasks HPO (Mem) | $82.18_{\pm0.26}$ | $94.75_{\pm0.28}$ | $56.94_{\pm0.66}$ | $83.08_{\pm0.21}$ |

## 5.1 RESULTS

For the split-task setting, the results of our experiments show that none of the HPO frameworks looked at perform much better than the rest. The results are presented in Table 2 and 3 and we have bolded the results which are better by $+0.5\%$ than any of the other HPO frameworks results for a given CL method. The reason we chose to bold results in this way is to be able to draw attention to and reference observed effect sizes. We want to do this because if the observed effect sizes are small it suggests that no method performs much better than any other and hence that other factors become more important when selecting a HPO framework, e.g. compute cost. In Table 2 there are few bolded results and for those that exist, the HPO framework which achieves it varies. This shows that, for the datasets shown in Table 2, there is only a small difference in performance between HPO frameworks. While in Table 3 there are more bolded results indicating a slightly greater variance in the performance of HPO frameworks—perhaps due to the greater complexity of the datasets looked at. However, as in Table 2, in Table 3 the HPO framework that performs the best differs across datasets and CL methods. These results show that no HPO framework performs consistently better than the rest. For instance, on CIFAR-100, no HPO framework improves accuracy over the other methods by more than $+0.5\%$ for all CL methods but ESMER in class-incremental learning. This suggest that for the split-task setting there is no general advantage in using one HPO framework over another in terms of predictive performance.

Table 3: Results of using different HPO frameworks for ER, iCaRL, ER-ACE, ESMER and DER++ on the standard split-task CORe50 and Tiny ImageNet benchmarks. We report mean average accuracy over three runs with their standard errors and, to highlight effect size, bold results which are greater by $+0.5\%$ average accuracy than any other for that CL method. The table shows that all HPO frameworks perform similarly; none perform consistently better than the rest.

| | | CORe50 | | Tiny ImageNet | |
|---|---|---|---|---|---|
| CL Method | HPO Framework | Class-IL. | Task-IL. | Class-IL. | Task-IL. |
| | End-of-training HPO | $37.37_{\pm1.03}$ | $55.51_{\pm0.41}$ | $28.01_{\pm0.09}$ | $68.17_{\pm0.06}$ |
| | First-task HPO | $38.37_{\pm0.38}$ | $56.95_{\pm0.62}$ | $28.51_{\pm0.18}$ | $\mathbf{68.72}_{\pm0.13}$ |
| ER | Current-task HPO | $35.97_{\pm0.24}$ | $53.40_{\pm1.01}$ | $25.79_{\pm0.21}$ | $66.96_{\pm0.15}$ |
| | Seen-tasks HPO (Val) | $\mathbf{39.12}_{\pm0.64}$ | $57.32_{\pm0.63}$ | $28.45_{\pm0.28}$ | $68.16_{\pm0.26}$ |
| | Seen-tasks HPO (Mem) | $36.10_{\pm1.15}$ | $54.28_{\pm0.77}$ | $\mathbf{29.58}_{\pm0.25}$ | $68.02_{\pm0.14}$ |
| | End-of-training HPO | $54.30_{\pm0.36}$ | $85.74_{\pm0.45}$ | $37.09_{\pm0.27}$ | $70.37_{\pm0.36}$ |
| | First-task HPO | $52.56_{\pm0.10}$ | $84.60_{\pm0.09}$ | $36.42_{\pm0.22}$ | $70.11_{\pm0.13}$ |
| iCaRL | Current-task HPO | $54.26_{\pm0.02}$ | $85.74_{\pm0.06}$ | $37.17_{\pm0.28}$ | $70.67_{\pm0.03}$ |
| | Seen-tasks HPO (Val) | $51.89_{\pm0.39}$ | $84.02_{\pm0.68}$ | $34.81_{\pm0.42}$ | $68.42_{\pm0.41}$ |
| | Seen-tasks HPO (Mem) | $49.16_{\pm0.23}$ | $82.43_{\pm0.23}$ | $36.79_{\pm0.13}$ | $70.46_{\pm0.08}$ |
| | End-of-training HPO | $39.33_{\pm0.79}$ | $58.14_{\pm1.29}$ | $\mathbf{38.94}_{\pm0.47}$ | $\mathbf{70.18}_{\pm0.23}$ |
| | First-task HPO | $37.81_{\pm0.71}$ | $56.02_{\pm0.60}$ | $36.94_{\pm0.67}$ | $68.16_{\pm0.30}$ |
| ER-ACE | Current-task HPO | $43.59_{\pm0.09}$ | $61.33_{\pm0.33}$ | $37.63_{\pm0.38}$ | $68.25_{\pm0.41}$ |
| | Seen-tasks HPO (Val) | $\mathbf{44.32}_{\pm0.69}$ | $\mathbf{62.28}_{\pm0.51}$ | $36.06_{\pm0.37}$ | $67.69_{\pm0.26}$ |
| | Seen-tasks HPO (Mem) | $37.60_{\pm0.69}$ | $56.01_{\pm1.17}$ | $32.37_{\pm0.34}$ | $64.37_{\pm0.47}$ |
| | End-of-training HPO | $45.08_{\pm1.06}$ | $62.05_{\pm0.45}$ | $\mathbf{47.33}_{\pm0.30}$ | $76.18_{\pm0.22}$ |
| | First-task HPO | $\mathbf{47.07}_{\pm1.18}$ | $63.69_{\pm0.95}$ | $46.69_{\pm0.56}$ | $75.72_{\pm0.24}$ |
| ESMER | Current-task HPO | $46.01_{\pm0.90}$ | $63.32_{\pm0.59}$ | $45.20_{\pm0.53}$ | $74.93_{\pm0.29}$ |
| | Seen-tasks HPO (Val) | $43.29_{\pm1.11}$ | $60.77_{\pm0.80}$ | $44.82_{\pm0.16}$ | $74.27_{\pm0.11}$ |
| | Seen-tasks HPO (Mem) | $42.15_{\pm1.24}$ | $58.78_{\pm1.10}$ | $44.26_{\pm0.20}$ | $74.54_{\pm0.31}$ |
| | End-of-training HPO | $51.87_{\pm0.44}$ | $63.48_{\pm0.61}$ | $\mathbf{39.89}_{\pm0.27}$ | $\mathbf{70.41}_{\pm0.17}$ |
| | First-task HPO | $46.07_{\pm1.58}$ | $58.07_{\pm1.18}$ | $35.98_{\pm0.63}$ | $65.86_{\pm0.37}$ |
| DER++ | Current-task HPO | $51.58_{\pm0.77}$ | $\mathbf{64.19}_{\pm046}$ | $36.64_{\pm0.33}$ | $66.43_{\pm0.49}$ |
| | Seen-tasks HPO (Val) | $49.19_{\pm0.37}$ | $62.10_{\pm0.65}$ | $31.88_{\pm5.36}$ | $64.20_{\pm3.00}$ |
| | Seen-tasks HPO (Mem) | $41.08_{\pm1.91}$ | $54.73_{\pm2.16}$ | $33.54_{\pm0.13}$ | $63.68_{\pm0.17}$ |

In the heterogeneous task setting we also see that none of the HPO frameworks perform consistently better than the rest. The results for this setting are presented in Table 4 and we have again bolded the results which are better by $+0.5\%$ than any of the other HPO frameworks for a given CL method. Like the results for the split-task setting, there are many columns for each CL method which have no bolded result and for the three which do the HPO framework which achieves it is different. Therefore, we conclude that in the heterogeneous task setting it is also the case that there is no one best HPO framework. The reason we look at the heterogeneous task setting is because we expected a greater benefit from adapting hyperparameters per task, given that unlike the split-task setting each task is quite different. However, our results show that this is not the case and that it is possible to use the same hyperparameters across all the tasks and still perform well.

**Performance of first-task HPO** Our results show that all of the HPO frameworks tested perform similarly. Therefore, we conclude that other factors should be used when deciding what realistic HPO framework to use on these common CL benchmarks. For example, taking computational cost into account would mean that first-task HPO would be a good method to use as it is the most computationally efficient. Given this, we describe here in more detail its relative performance compared to the other HPO frameworks tested. In the split-task setting, we see from Table 2 and 3, that for ER some of its results are bolded. Thus, first-task HPO sometimes achieves the best performance. Additionally, for the spilt task setting, there is an average performance difference from end-of-training HPO to first-task HPO of $-0.62\%$ in class-incremental learning and $-0.91\%$ in task-incremental learning. While, for the heterogeneous tasks setting there is an average performance difference from

Table 4: Results of using different HPO frameworks for ER, iCaRL, ER-ACE, ESMER and DER++ on heterogeneous task benchmarks. We report mean average accuracy over three runs with their standard errors and, to highlight effect size, bold the results which are greater by $+0.5\%$ accuracy than any other for that CL method. The table shows that no HPO framework is consistently better than the rest.

| CL Method | HPO Framework | Hetero-CIFAR-100 Class-IL. | Hetero-TinyImg Class-IL. |
|---|---|---|---|
| ER | End-of-training HPO | $50.41_{\pm 0.21}$ | $39.41_{\pm 0.57}$ |
| | First-task HPO | $50.33_{\pm 0.50}$ | $40.77_{\pm 0.34}$ |
| | Current-task HPO | $49.77_{\pm 0.21}$ | $40.65_{\pm 0.97}$ |
| | Seen-tasks HPO (Val) | $\mathbf{51.70}_{\pm 0.23}$ | $40.55_{\pm 0.22}$ |
| | Seen-tasks HPO (Mem) | $45.52_{\pm 0.41}$ | $\mathbf{44.62}_{\pm 0.18}$ |
| iCaRL | End-of-training HPO | $51.54_{\pm 0.38}$ | $37.17_{\pm 0.48}$ |
| | First-task HPO | $49.81_{\pm 0.10}$ | $37.47_{\pm 0.26}$ |
| | Current-task HPO | $51.34_{\pm 0.32}$ | $37.07_{\pm 0.07}$ |
| | Seen-tasks HPO (Val) | $48.15_{\pm 0.09}$ | $35.70_{\pm 0.23}$ |
| | Seen-tasks HPO (Mem) | $47.87_{\pm 0.15}$ | $35.27_{\pm 1.12}$ |
| ER-ACE | End-of-training HPO | $51.96_{\pm 0.60}$ | $\mathbf{45.47}_{\pm 0.42}$ |
| | First-task HPO | $51.37_{\pm 0.16}$ | $43.62_{\pm 1.09}$ |
| | Current-task HPO | $51.78_{\pm 0.30}$ | $43.87_{\pm 0.20}$ |
| | Seen-tasks HPO (Val) | $51.94_{\pm 0.12}$ | $43.15_{\pm 0.63}$ |
| | Seen-tasks HPO (Mem) | $48.15_{\pm 0.28}$ | $42.19_{\pm 0.84}$ |
| ESMER | End-of-training HPO | $50.54_{\pm 0.16}$ | $44.87_{\pm 0.26}$ |
| | First-task HPO | $50.43_{\pm 0.34}$ | $45.84_{\pm 0.50}$ |
| | Current-task HPO | $50.68_{\pm 0.31}$ | $44.50_{\pm 0.31}$ |
| | Seen-tasks HPO (Val) | $47.96_{\pm 0.61}$ | $42.18_{\pm 0.22}$ |
| | Seen-tasks HPO (Mem) | $50.56_{\pm 0.40}$ | $46.00_{\pm 0.43}$ |
| DER++ | End-of-training HPO | $54.12_{\pm 0.70}$ | $46.41_{\pm 0.77}$ |
| | First-task HPO | $54.87_{\pm 0.39}$ | $43.45_{\pm 3.55}$ |
| | Current-task HPO | $55.10_{\pm 0.52}$ | $45.95_{\pm 0.93}$ |
| | Seen-tasks HPO (Val) | $54.67_{\pm 0.57}$ | $46.51_{\pm 0.49}$ |
| | Seen-tasks HPO (Mem) | $49.06_{\pm 3.90}$ | $25.78_{\pm 7.40}$ |

end-of-training HPO to first-task HPO of $-0.39\%$. These results indicate, compared to standard practice, that by using first-task framework it is possible to perform realistic HPO for much less computation with only a small expected cost to performance. However, it is important to point out that first-task HPO has a failure case of when the first task is not informative for the hyperparameter choices of subsequent tasks. This failure case does not happen on the standard CL benchmarks used in this work nor in the heterogeneous task setting where the tasks are designed to be more different. Therefore, it is an open question whether such a failure case will arise if the standard CL benchmarks used by the community change to be different, hopefully more realistic, data streams.

One of the potential reasons that the performance is similar between HPO frameworks is that there is little variation between the performance of different hyperparameter configurations. To see whether this is the case, we have plotted in Figure 3 histograms of the performance of using different fixed HPO configurations for DER++. The histograms show that hyperparameter configurations achieve a wide range of average accuracies. Therefore, the performance of different HPO configurations is *not* the reason why the HPO frameworks have similar results. Additionally, in Appendix B, we examine whether using default hyperparameters performs as well as selecting hyperparameters using HPO. We found that using default hyperparameters in most cases performed worse than using a HPO framework. Hence, our results suggest that HPO is necessary but that out of the HPO frameworks tested there is no one best performing method.

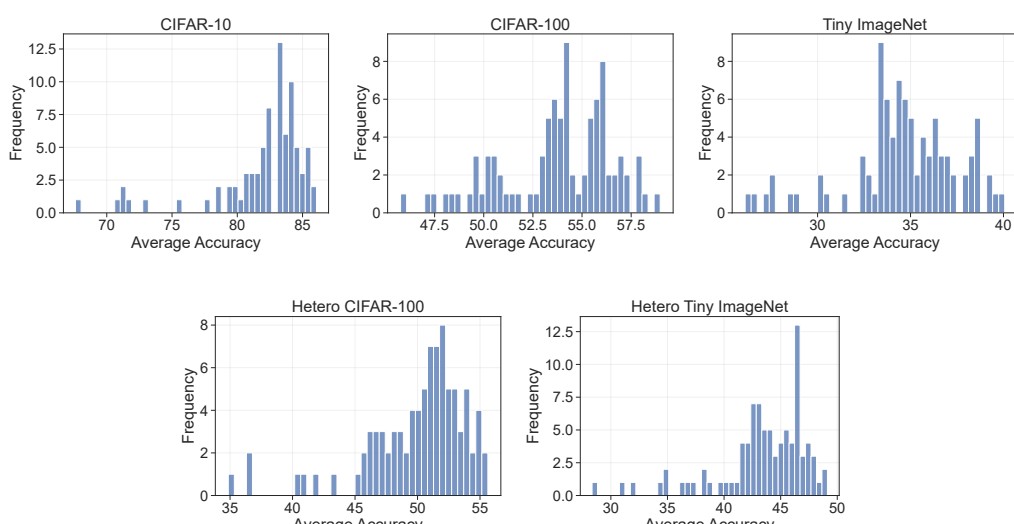

Figure 3: Histograms of the validation accuracy at the end of training for each hyperparameter setting searched over for DER++. We look at standard CL benchmarks and heterogeneous task benchmarks, which are identified by having a 'Hetero' in their name. The histograms show that different hyperparameter settings give a varying range of performances and only a few achieve near to the top performance.

## 6  CONCLUSIONS

In this paper we have benchmarked several hyperparameter optimisation (HPO) frameworks for CL which are more realistic than the currently commonly used end-of-training HPO framework. We benchmarked both fixed HPO frameworks, which fix the hyperparameters throughout training, and dynamic HPO frameworks that continually adapt the hyperparameters. Our results show for commonly used CL benchmarks that all the HPO frameworks achieve similar performances and none consistently outperforms the others. Because of this, we recommend that practitioners using these benchmarks should select a realistic HPO framework using other factors—for example compute cost, for which *first-task* HPO is a good choice. Our results also suggest that future work on HPO for CL should move towards the use of new benchmarks where a difference in performances across HPO frameworks could arise.

### REPRODUCIBILITY STATEMENT

To make our experiments reproducible, we provide in Section 5 a description of the setup used in this work, which is in common with many other works in CL (Buzzega et al., 2020), and provide more specific experimental details in Appendix A. Additionally, we provide the code used in the supplementary material.

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

## A   ADDITIONAL EXPERIMENTAL DETAILS

While we have aimed to include all the main experimental details in the main paper there are a few others to mention here. First, we mostly follow the experimental setup of Buzzega et al. (2020) and Boschini et al. (2022) and use the Mammoth library produced by those works as the base of our code. Second, we use as our optimiser SGD with no momentum or weigh decay, as is done in other works (Aljundi et al., 2019b; Buzzega et al., 2020; Chaudhry et al., 2019a; Lee & Storkey, 2023). Third, in the heterogeneous tasks setting we look at tasks sequences where each task in order has the following number of classes associated with it $[9, 2, 7, 3, 4, 9, 8, 3, 3, 7, 4, 4, 5, 9, 4, 5, 2, 8, 2, 2]$ and all the data of a class is contained in the task associated with it. For Tiny ImageNet we only use the first 100 classes in the heterogeneous tasks setting to reduce runtime and to make it more comparable to CIFAR-100 in that setting. In the heterogeneous tasks setting each task has a variable amount of data. For example, using CIFAR-100, the first task contains nine classes and so it will contain in total $4500$ examples ($500$ examples per task) while the second task contains two classes so will only contain 1000 examples. Also, as in each task the learner needs to discriminate between a varying number of classes the difficultly should vary between tasks. Additionally, in the heterogeneous tasks setting we only look at class-incremental learning. Finally, the CORe50 dataset consists of data drawn from multiple different background and lighting environments called *sessions* and the test data consists of data from different sessions than the training data. Therefore, to insure that we more accurately model the covariate shift from the training to test data in our validation signal, we construct the validation sets for CORe50 differently from the other datasets where it is sampled randomly. Specifically, we use the data of *session 2* contained in the task as the validation data for that task.

We record here the hyperparameter grid that we sample over when performing HPO. We look at learning rates in the set $\{0.2, 0.15, 0.1, 0.075, 0.05, 0.03, 0.01, 0.0075, 0.005, 0.0025\}$. For DER++, we perform HPO over both regularisation coefficients where we sample $\alpha$ in the set $\{0.2, 0.5, 1.0\}$ and $\beta$ in the set $\{0.2, 0.5, 1.0\}$. For ESMER, we perform HPO over the loss margin coefficient where we sample over the set $\{1.5, 1.2, 1.0\}$. We sample all possible combinations of learning rates and regularisation coefficients in each of our HPO frameworks. This grid contains the ones used in the popular works Buzzega et al. (2020), Boschini et al. (2022) and Sarfraz et al. (2023), where we add additional learning rate settings and, for some datasets, regularisation coefficients settings. We note here that while we use grid search in this paper to align with common practice in CL (Buzzega et al., 2020; Delange et al., 2021), any hyperparameter sampling/selecting method can be used with each of the HPO frameworks looked at. For example, tree-structured Parzen estimators are a common Bayesian HPO method to sample hyperparameter configurations for neural networks (Bergstra et al., 2011). Additionally, Gaussian process based HPO methods are also commonly used (Snoek et al., 2012) and have been looked at in settings related to online learning (Hellan et al., 2023).

## B   EXPERIMENTS USING DEFAULT HYPERPARAMETER VALUES

To test whether HPO is needed in CL and if instead using default hyperparameters is sufficient, we perform experiments using default hyperparameters. The experimental setup is the same as the main paper and we use for the default learning rate the default given by PyTorch, $0.001$, and use $1.0$ as the default for regularisation coefficients. The results are presented in Tables 5 and 7. The tables show that using default hyperparameters leads to worse performance than using HPO. Additionally, for some dataset and CL method combinations the default hyperparameters perform very badly showing the need to adapt hyperparameters to the dataset and CL method used.

Table 5: Comparison of using default hyperparameters versus using a HPO framework on split-task CIFAR-10 and CIFAR-100, where we only present the most common HPO framework (End-of-training HPO) and the most efficient (First-task HPO) for readability. We report mean average accuracies over three runs with their standard errors. The table shows that using default HPs leads to worse performance than using HPO for standard CL benchmarks.

| CL Method | HPO Framework | CIFAR-10 | | CIFAR-100 | |
|---|---|---|---|---|---|
| | | Class-IL. | Task-IL. | Class-IL. | Task-IL. |
| ER | End-of-training HPO | $83.55_{\pm0.44}$ | $97.18_{\pm0.14}$ | $51.03_{\pm0.43}$ | $85.68_{\pm0.29}$ |
| | First-task HPO | $84.38_{\pm0.45}$ | $96.82_{\pm0.17}$ | $49.61_{\pm0.34}$ | $84.97_{\pm0.19}$ |
| | Default HPs | $74.60_{\pm0.79}$ | $94.53_{\pm0.13}$ | $35.39_{\pm0.36}$ | $72.83_{\pm0.24}$ |
| iCaRL | End-of-training HPO | $77.79_{\pm0.23}$ | $98.52_{\pm0.03}$ | $54.30_{\pm0.36}$ | $85.74_{\pm0.45}$ |
| | First-task HPO | $77.83_{\pm0.22}$ | $95.31_{\pm0.12}$ | $52.56_{\pm0.10}$ | $84.60_{\pm0.09}$ |
| | Default HPs | $68.34_{\pm0.49}$ | $92.98_{\pm0.21}$ | $11.54_{\pm0.25}$ | $41.66_{\pm0.54}$ |
| ER-ACE | End-of-training HPO | $82.34_{\pm0.30}$ | $96.74_{\pm0.01}$ | $55.58_{\pm0.39}$ | $85.73_{\pm0.09}$ |
| | First-task HPO | $83.20_{\pm0.79}$ | $96.67_{\pm0.18}$ | $56.36_{\pm0.29}$ | $86.11_{\pm0.154}$ |
| | Default HPs | $75.46_{\pm0.21}$ | $94.71_{\pm0.06}$ | $42.65_{\pm0.57}$ | $76.28_{\pm0.19}$ |
| ESMER | End-of-training HPO | $80.73_{\pm0.15}$ | $96.50_{\pm0.01}$ | $56.16_{\pm0.54}$ | $88.69_{\pm0.35}$ |
| | First-task HPO | $77.89_{\pm0.46}$ | $96.15_{\pm0.12}$ | $56.61_{\pm0.20}$ | $89.05_{\pm0.10}$ |
| | Default HPs | $68.86_{\pm1.06}$ | $93.54_{\pm0.20}$ | $42.94_{\pm0.61}$ | $79.64_{\pm0.36}$ |
| DER++ | End-of-training HPO | $84.40_{\pm0.94}$ | $95.75_{\pm0.33}$ | $56.04_{\pm3.67}$ | $83.13_{\pm2.69}$ |
| | First-task HPO | $85.22_{\pm0.08}$ | $96.14_{\pm0.10}$ | $55.20_{\pm0.78}$ | $81.68_{\pm0.66}$ |
| | Default HPs | $77.59_{\pm0.45}$ | $93.83_{\pm0.40}$ | $46.11_{\pm1.16}$ | $78.14_{\pm1.28}$ |

Table 6: Comparison of using default hyperparameters versus using a HPO framework on split-task Tiny ImageNet, where we only present the most common HPO framework (End-of-training HPO) and the most efficient (First-task HPO) for readability. We report mean average accuracies over three runs with their standard errors. The table shows that using default HPs leads to worse performance than using HPO for standard CL benchmarks.

| CL Method | HPO Framework | TinyImageNet | |
|---|---|---|---|
| | | Class-IL. | Task-IL. |
| ER | End-of-training HPO | $28.01_{\pm0.09}$ | $68.17_{\pm0.06}$ |
| | First-task HPO | $28.51_{\pm0.18}$ | $68.72_{\pm0.13}$ |
| | Default HPs | $16.27_{\pm0.20}$ | $50.99_{\pm0.41}$ |
| iCaRL | End-of-training HPO | $37.09_{\pm0.27}$ | $70.37_{\pm0.36}$ |
| | First-task HPO | $36.42_{\pm0.22}$ | $70.11_{\pm0.13}$ |
| | Default HPs | $5.30_{\pm0.03}$ | $23.97_{\pm0.10}$ |
| ER-ACE | End-of-training HPO | $38.94_{\pm0.47}$ | $70.18_{\pm0.23}$ |
| | First-task HPO | $36.94_{\pm0.67}$ | $68.16_{\pm0.30}$ |
| | Default HPs | $25.84_{\pm0.26}$ | $56.25_{\pm0.13}$ |
| ESMER | End-of-training HPO | $47.33_{\pm0.30}$ | $76.18_{\pm0.22}$ |
| | First-task HPO | $46.69_{\pm0.56}$ | $75.72_{\pm0.24}$ |
| | Default HPs | $33.11_{\pm0.39}$ | $63.15_{\pm0.17}$ |
| DER++ | End-of-training HPO | $39.89_{\pm0.27}$ | $70.41_{\pm0.17}$ |
| | First-task HPO | $35.98_{\pm0.63}$ | $65.86_{\pm0.37}$ |
| | Default HPs | $25.66_{\pm0.16}$ | $59.14_{\pm0.51}$ |

Table 7: Comparison of using default hyperparameters versus using a HPO framework on heterogeneous task benchmarks, where we only present the most common HPO framework (End-of-training HPO) and the most efficient (First-task HPO) for readability. We report mean average accuracies over three runs with their standard errors. The table shows that using default HPs leads to worse performance than using HPO for heterogeneous task benchmarks.

| CL Method | HPO Framework | Hetero-CIFAR-100 Class-IL. | Hetero-TinyImg Class-IL. |
|---|---|---|---|
| ER | End-of-training HPO | $50.41_{\pm 0.21}$ | $39.41_{\pm 0.57}$ |
|  | First-task HPO | $50.33_{\pm 0.50}$ | $40.77_{\pm 0.34}$ |
|  | Default HPs | $33.76_{\pm 0.78}$ | $26.88_{\pm 0.45}$ |
| iCaRL | End-of-training HPO | $51.54_{\pm 0.38}$ | $37.17_{\pm 0.48}$ |
|  | First-task HPO | $49.81_{\pm 0.10}$ | $37.47_{\pm 0.26}$ |
|  | Default HPs | $12.23_{\pm 0.19}$ | $10.6_{\pm 0.26}$ |
| ER-ACE | End-of-training HPO | $51.96_{\pm 0.60}$ | $45.47_{\pm 0.42}$ |
|  | First-task HPO | $51.37_{\pm 0.16}$ | $43.62_{\pm 1.09}$ |
|  | Default HPs | $38.11_{\pm 0.80}$ | $32.37_{\pm 0.53}$ |
| ESMER | End-of-training HPO | $50.54_{\pm 0.16}$ | $44.87_{\pm 0.26}$ |
|  | First-task HPO | $50.43_{\pm 0.34}$ | $45.84_{\pm 0.50}$ |
|  | Default HPs | $37.92_{\pm 0.30}$ | $34.22_{\pm 0.41}$ |
| DER++ | End-of-training HPO | $54.12_{\pm 0.70}$ | $46.41_{\pm 0.77}$ |
|  | First-task HPO | $54.87_{\pm 0.39}$ | $43.45_{\pm 3.55}$ |
|  | Default HPs | $44.43_{\pm 0.51}$ | $30.21_{\pm 1.53}$ |

