# OpenReview forum: "Towards Realistic Hyperparameter Optimization in Continual Learning"
_ICLR.cc/2025/Conference — ICLR 2025 Conference Withdrawn Submission_

### Official Review · Reviewer_A6nR · 2024-10-31

**Soundness:** 2
**Presentation:** 3
**Contribution:** 2
**Rating:** 3
**Confidence:** 5

**Summary:**

The paper explores the impact of different strategies to assemble validation sets for hyperparameter optimization (HPO) purposes in continual learning (CL) scenarios. Different strategies are benchmarked on a selection of CL methods from the experience replay family. The benchmarking study shows that no HPO strategy offers a distinctive advantage over the other on the selected CL methods, on the target hyperparameters and based on the chosen CL datasets. The main takeaway message of the work is that since different strategies do not yield to substantially different predictive accuracies, then one should select the HPO approach based on other factors, such as efficiency.

**Strengths:**

* The research question behind the paper is a significant one: HPO is a largely overlooked aspect in CL and it is typically addressed with either unrealistic approaches looking at end-of-training performance or with approaches whose computational costs are substantially in contrast with the very reasons for using CL (e.g. such as in current-task HPO).

* The work has some originality in that, to the extent of my knowledge, this is a first attempt at a structured benchmarking of different  strategies to build validation sets in HPO for CL.

* If one accepts the restrictions of the scenario (elaborated in the weaknesses), the benchmarking assessment is executed in a fairly robust way, with multiple replay strategies being considered, assessing the effects of tasks of varying complexity and reporting baseline performance for the default hyperparameters.

* Quality of the writing and structuring of the paper is good.

**Weaknesses:**

* The work is not really about an assessment of HPO in CL. The only HPO strategy that is considered here is grid search. What is being assessed is how different ways in which you can pick up the validation set(s) as part of datasets received continual-style, influence the predictive performance of CL strategies whose hyperparameters are not selected having a continual HPO setting in mind. Continual HPO is actually about how you perform positive information transfer between tasks at the level of the hyperparameter adaptation strategy, without resorting to inefficient grid-like search and while catering for knowledge retention between tasks. Which is pretty much the HPO equivalent of CL for parameter update. This choice is evident from the fact that while the paper reference actual attempts at performing continual HPO, it does not benchmark any of them. The sheer effect of this is that there is little "realism" in all the tested strategies, in CL sense.

 * An additional limit to the claimed generality of the study is that the benchmarking exercise is not about CL. It is about a very specific family of methodologies, i.e. replay-based ones, which are arguably the ones whose predictive performance depends less on model-selection/HPO choices. Before coming down to the conclusions that existing benchmarks datasets are not sufficient to differentiate the effects of different HPO strategies, one would need to explore the effect on regularization-based and of architectural CL methodologies. Those are the ones which are likely to be heavily influenced by the choice of hyperparameters. Consider for instance the weight of the regularization terms to avoid forgetting, and their impact on the stiffening of the model ability to adapt its parameters on the current task. Even more so, if one considers architectural approaches where HPO may entail topological changes to the network which may render the adaptive model discontinous (at inference time) between task experiences (and hence strategies such as current-task HPO will loose meaning at all).

 *  The empirical analysis, while robust, focuses on very few hyperparameters and does not provide any insight into the differential effect of different hyperparameters (e.g. how much they tend to vary across experiences). Additionally, the empirical setting assumes partitioned classes. While being a common assumption, this is highly unrealistic, which contrasts with the claimed realism targeted by the work.

**Questions:**

1) Have you measured the effects of the different strategies on how different hyperparameters are chosen across experiences? Is there any hyperparameter that is more impacting?

2) Have you considered the effects of varying task ordering? Does this affect in any way the outcomes of increases the variance of results?

3) Do you have any evidence that the results in the work would generalize out of ER-based approaches?

---

### Official Review · Reviewer_s3Z5 · 2024-11-01

**Soundness:** 2
**Presentation:** 3
**Contribution:** 1
**Rating:** 3
**Confidence:** 5

**Summary:**

This work empirically evaluates different hyperparameter optimization (HPO) strategies for continual learning (CL). Using different CL methods and benchmarks, the authors conclude that all HPO strategies perform equally and that the strategy with lowest compute requirements should be chosen, i.e., tuning only on the first task and then keep the hyperparameter settings fixed.

**Strengths:**

The authors clearly describe their work as well as their motivation. The question of how to tune the hyperparameters in CL and it is great that the authors highlight that "end-of-training HPO" is no viable choice in practice, but the standard in CL.

**Weaknesses:**

The authors should have discussed in more detail the pros and cons for the different HPO strategies. In the following some ideas what could be discussed or my opinion on each of the strategies:
- first-task HPO: many CL methods, e.g., DER++, have hyperparameters that only matter if there is more than one task. how can you tune them in this case?
- current-task HPO: this is useless since it does not allow to measure for important metrics such as forgetting.
- seen-tasks HPO: this increases the required memory size since this is data we cannot train on. some scenarios don't allow for any memory. is it better to use the additional data for HPO or for training?

Besides the clear disadvantages of the methods discussed above, we don't observe this empirically.

The authors imply that we care about "end-of-training" performance. They completely overlook that we actually care about anytime performance and that the concept of "end" doesn't exist in CL. It would be great if the authors would show how performance changes over time. Besides this fact, it's a common plot in many CL papers as well as a nice ablation study.

What about other metrics such as forgetting? As I've mentioned above, some HPO strategies would completely ignore it and we'd assume to see this in this metric.

The authors argue that all tested methods perform equally and therefore "first-task HPO" should be preferred since it is computationally cheap. In my opinion such a recommendation should not be made. The authors should make clear when exactly this choice is justified.

**Questions:**

is it worth using seen-tasks HPO? would it be better to use the validation data as part of training instead?

---

### Official Review · Reviewer_85a3 · 2024-11-01

**Soundness:** 3
**Presentation:** 3
**Contribution:** 1
**Rating:** 3
**Confidence:** 3

**Summary:**

The paper illuminates the challenges in hyper parameter optimization and tries to tell us which HPS framework is better for CL.

**Strengths:**

Presentation:
	The paper presents things clearly and the ideas are understandable.

Technical Correctness
	  The paper appears correct in all aspects and the main conclusions are intuitive and very nice. I do have some issues with the conclusion that we should pick frameworks based on the computational complexity of the paper, I understand that most HPO methods behave the same with very little difference in performance.

**Weaknesses:**

The novelty of this work is questionable. The paper surveys the different HPS approaches in the CL setting and describes the limitations and challenges of training CL with HPO. This is most likely a benchmark and datasets papers submitted to the main track of ICLR. The key contribution of this work is the code, that can be used to test further and novel HPO methods in the CL paradigm.

**Questions:**

How do different continual learning settings (e.g., task-incremental vs. class-incremental learning) impact the performance of various HPO frameworks?

What type of HPO mechanisms have been utilized for training?

Since computational efficiency is not always considered the best metric for verifying the goodness of a CL method. Can the authors illuminate under what conditions we should be talking about computational efficiency and under what conditions we should be thinking about performance.

---

### Official Review · Reviewer_NyMH · 2024-11-04

**Soundness:** 2
**Presentation:** 2
**Contribution:** 2
**Rating:** 3
**Confidence:** 4

**Summary:**

This paper benchmarks various hyperparameter optimization (HPO) frameworks in continual learning (CL). The paper argues that none of the compared frameworks seems to consistently outperform the other.

**Strengths:**

- The paper correctly recognizes that HPO is an open problem in the CL literature.
- The paper motivates very well why it is worthwhile to study the HPO problem in CL.

**Weaknesses:**

- The paper claims that none of the compared HPO methods seems to be superior to the others. I found it a bit disappointing that there was no further examination of why that was the case. Does it really make no difference (in terms of learning performance) which HPO method one uses, or could it be that certain HPO methods are better suited to CL methods with some particular characteristics?
- The benchmark contains just five CL methods, and the paper does not really explain why these particular methods were selected.
- I think three runs are too few to have a relatively accurate picture of how the various HPO methods compare to each other.
- I think the paper would benefit from a short section on its limitations. For instance, there are online CL settings and CL settings without crisp transitions between tasks, which might not be compatible with the CL definition followed within the paper. Also, the experiments are performed on only one data modality (images) and with only one neural network architecture.
- I think that having access to validation sets from previous tasks is arguably an unrealistic assumtpion, not only for the reason given in the paper, but also because it might be more beneficial to actually use the data in these sets during training instead of just using them as validation sets.
- How did you decide on the value of 0.5%, which you use to decide which table entries to write in bold.

**Questions:**

- I think the term end-of-training HPO might be confusing, because it incorrectly implies that the HPO takes place at the end of training. Also, it is not mandatory to use the accuracy at the end of training to decide on the optimal HP configuration. One could also use other metrics such as forgetting which are not only dependent on learning performance at the end of training. My suggestion would be to use a term that refers to the actual HPO process followed (i.e., repeated training on the full stream). What do you think?
- In Seen-tasks HPO (Val) the combined validation set probably contains data of each class in roughly equal proportions. I'm assuming that that is not the case in Seen-tasks HPO (Mem), which can make the HPO more sensitive to the majority classes. Did you take that into account in any way?
- Why is the heterogeneous task setting more computationally costly than the standard split-task setting? (See Line 297 "We only look at CIFAR-100 and Tiny ImageNet for the heterogeneous task setting due to computational cost.")

---

### Note · Authors · 2024-11-23

I have read and agree with the venue's withdrawal policy on behalf of myself and my co-authors.